# OpenReview forum: "FAIRMINDSIM: ALIGNMENT OF BEHAVIOR, EMO- TION, AND BELIEF IN HUMANS AND LLM AGENTS AMID ETHICAL DILEMMAS"
_ICLR.cc/2025/Conference — Submitted to ICLR 2025_

### Official Review · Reviewer_o9ka · 2024-10-25

**Soundness:** 2
**Presentation:** 2
**Contribution:** 2
**Rating:** 3
**Confidence:** 3

**Summary:**

The paper presents FarMindSim, a framework designed to simulate moral dilemmas to explore AI alignment with human ethical and social values. It investigates how embedded human personality traits in large language models (LLMs) affect their decision-making in moral scenarios, particularly focusing on altruistic punishment in a structured economic game. The scenario involves three players: player 1 makes an unfair distribution of a payout sum between itself and a static player 2, while player 3 (portrayed by an LLM or human) can intervene at a personal cost to change this distribution. The study introduces the Belief-Reward Alignment Behavior Evolution Model (BREM), an extension of the Recursive Reward Model (RRM), to analyze how intrinsic beliefs and rewards impact decision-making. The findings suggest that intrinsic values hold a more significant influence than (payout) rewards on decision-making in both humans and LLMs.

**Strengths:**

The paper addresses a compelling intersection of sociology and AI by examining how social and emotional traits influence decision-making in both LLMs and humans. Furthermore, there a some intriguing findings such as the conclusion that LLMs are less influenced by emotions compared to humans, which seems intuitive and well-supported. The proposed BREM model for assessing value-drive decision-making in LLMs is a conceptually interesting and novel contribution.

**Weaknesses:**

The paper is of an experimental nature, and unfortunately this is exactly where it, in my opinion, lacks depth. The main focus is on a single family of LLMs (OpenAI) and one specific game environment, which limits the generalizability of the findings. Expanding the study to include other LLM families (e.g., Llama, Claude, Gemini, etc.) and more diverse game scenarios (e.g., that likewise exhibit the altruistic punishment phenomena, in a different setting) would add significant depth and weight to the reported findings (please also see my questions). Furthermore, under the assumptions that LLMs have an intrinsic set of values, I would expect that these emerge from various fine-tuning procedures that differ between models, which further corroborates the need to include different families in the experiments. In summary, the results provided in the paper, while insightful, feel incomplete without broader comparisons across different LLM families and scenarios.

**Questions:**

## General Questions
1. What is the rationale behind the math problems posed to the two players? Are they always of equal difficulty? Is there any variability in the problems, and does this affect the experiment’s outcomes? The paper mentions “*In Stage 1, Player One and Player Two each solves three simple math problems; if both answer correctly, they jointly receive a reward of 3 RMB.*” — are trials with incorrect answers discarded (please see also my question below)?
2. Have you considered how problem difficulty or correctness of the players’ solutions impacts the third player? Would an unjust distribution of money be harder to justify if one player performed poorly compared to the other? For example, I would imagine that a human would be more inclined to intervene if player 1 failed to solved its task, whereas player 2 solved it correctly, and yet it got less reward than player 1.
3. Could you explain why you chose to limit the study to one family of LLMs (OpenAI)? Do you perhaps already have some intuition on how would presented results compare across different LLM families? How about different fine-tuning strategies, do you have an intuition on how would methods such as Constitutional AI and RLHF compare?
4. Are there more examples from economics games that you have initially considered for your experiments, perhaps those that likewise aim to exhibit the altruistic punishment concept from its participants? Do you think results reported in the paper would translate also to those different scenarios?
5. Can you provide more details about the three modules constituting the agent? Specifically, why these three were necessary, and how they compare to simpler, prompt-based approaches? The memory module explanation (line 188) particularly seems underdeveloped.

## Minor Improvements and Comments

- Line 75: The sentence seems out of place;
- Line 86: Typo in “thedifferences.”;
- Line 89: Did you mean “BREM” instead of “EREM”?
- Line 152: "llm" -> "LLM";
- Line 155: "…which…" -> "…in which…";
- Line 257: Please clarify whether “different experiment” should be “different conditions”, as mixing the terms may cause confusion;
- Line 424: Typo;

---

> ### Author Response · Authors · 2024-11-23
>
> Thank you very much for your careful reading and very insightful comments. We offer detailed responses to your comments below.
> ### Q1 and Q2:
> To be precise, Player 1 and Player 2 are theoretically supposed to receive an equal distribution based on the game assumptions. However, in practice, the amount is distributed randomly and unfairly, as detailed in Table 2.
> ### Q3 and Q4:
> This study focuses on comparing the results of humans and the GPT series models to explore the differences in their performance within the field of psychology. The current experimental design and methodology are based on certain assumptions that require validation through previous related research to ensure scientific rigor and effectiveness. Additionally, the widespread application of GPT series models in the social sciences makes their inclusion in our study a logical choice. Ultimately, we aim for our research to provide interesting experimental results and analyses that contribute to both the AI and psychology fields, fostering the development of interdisciplinary research.
> ### Q5:
> The design here aims to make the dialogue logic of the LLM as similar as possible to that of humans, ensuring that the way the LLM makes decisions closely resembles human decision-making processes. To achieve this, it requires the integration of a Profiling Module, a Memory Module, and a Decision-Making Module.
>
> We appreciate the reviewer's thorough examination of our work and the identification of typographical errors, which we have corrected. Additionally, we have highlighted these corrections using different colors for clear distinction.

---

> ### Comment · Reviewer_o9ka · 2024-11-23
> **Thank you for your response**
>
> I thank the authors for their response. After consideration, I have decided to maintain my initial rating.

---

### Official Review · Reviewer_wqtJ · 2024-11-04

**Soundness:** 2
**Presentation:** 2
**Contribution:** 3
**Rating:** 3
**Confidence:** 4

**Summary:**

This paper proposes an evaluation scheme for measuring alignment of different decision-making entities with respect to the moral value of fairness. This scheme includes FairMindSim, a simulated suite of unfair scenarios where decision-making agents face the dilemma of undoing injustice by sacrificing a fixed amount of monetary value. The decision-making agents considered in this paper are humans and GPT models with diverse persona features. The second component of the proposed evaluation scheme is a reward model framework BREM, that aims to quantify the connection between beliefs (about fairness), decisions, and emotions (valence and arousal). Experiments conducted using this evaluation approach compare humans and GPT models across different aspects of fairness alignment.

**Strengths:**

I believe that the problem addressed in this paper is an interesting one, and that it was well-motivated by the authors.

I found FairMindSim to be a novel and promising approach for evaluating alignment with respect to moral values such as fairness. Extending this idea to other human values could result in a strong benchmark for measuring how close modern LLMs are to human values. One aspect to be careful for here is that LLMs should be evaluated on their understanding of human values instead of how well they memorize related patterns.

I found some findings of this paper to be quite interesting. For example, Section 4.2 provides some useful and comprehensive experimental results on how LLMs perceive their ''emotions'' when dealing with moral dilemmas, and how well these match the emotions reported by humans. This section is also well-backed by Section 3.1.

**Weaknesses:**

**Typos:** I usually do not complain about such things, but this paper has too many typos. In case the paper gets accepted, I would urge the authors to put some effort in correcting them.

**Related work section:** I found this section to be too lengthy and not particularly relevant to the content of the paper. It reads more like additional background on some loosely related concepts than actual related work, and hence it can be made more condensed or be moved to appendix (at least in part). What I would expect in this section instead is that authors mention studies that look into the problem at focus or related ones, and describe how these approaches compare to the one proposed in their paper. Could I respectfully ask the authors to provide such a paragraph in their rebuttal?

**Design choices for FairMindSim:** I found some design choices made in the proposed suite of simulated scenarios to be a bit unclear. More specifically, I am referring to the information individuals had to report regarding their persona. See my questions below.

**Sections 3.2, 4.3 and 4.4:** I had a hard time understanding Section 3.2. Most of the modeling choices here seem arbitrary to me. The only exception is the cumulative reward function which was clear. I understood what most of the measures introduced are aimed for, but why they were modeled like this was unclear to me. In an effort of enhancing my understanding of this part I have added questions regarding my main concerns below. As an immediate consequence of this, the merit of the experimental results presented in Sections 4.3 and 4.4 becomes questionable to me.

**Experimental results:** Some of the experimental results seemed a bit counter-intuitive and others were not adequately explained. See questions below.

**Questions:**

**Profiling module:** What is the reasoning behind the features you chose to constitute this module? Are they backed up by findings from prior work?

**Participants assignment:** Human participants are assigned to selfish and extreme selfish groups based on some metric or arbitrarily by the authors?

**CRF:** Are Cumulative and Cognitive Reward Functions, both abbreviated as CRF, the same thing?

**Sections 3.2:** Can you explain the following:
- what is behavior payoff in your BERM framework
- how do you quantify the level of unfairness E
- why Equation 2 is the correct way of formalizing the misalignment between beliefs and monetary incentives? Why is a linear relationship suitable in this case?
- what is the reasoning behind incorporating the influence of emotions in your framework as temperature in the BT model? Does this modeling choice stems from prior work or is motivated in any other way?
- what does it mean ''BDF backfires on beliefs'', can you explain the intuition of this update rule?

**Table 2:** What are the conditions you are referring to in Table 2? Selfish and extreme selfish or something else?

It seems that GPT-4o admits less CRF than humans, almost half. This seems a bit counter-intuitive. Are you suggesting that aligning with human values is not the same as aligning with human moral judgements?

Can you elaborate on the findings in lines 415-419. Do they suggest that there is a bias against gender in the LLMs you test for?

**Section 4.3:** You are mentioning that emotional factors have a significant impact on human beliefs. From your results however I understand that there is a correlation, without a clear causal arrow. Intuitively, I would expect the opposite direction from the one you are suggesting. Could you please elaborate on how you reached this conclusion?

---

> ### Author Response · Authors · 2024-11-23
>
> Thank you very much for your careful reading and very insightful comments. The related work section has been finalized, with the human-related content moved to the appendix as supplementary information. Regarding comparisons with existing methods, our approach is exploratory in nature. The comparison data in our study is derived from a specifically designed game, focusing on the comparison between humans and LLMs. In contrast, most mainstream methods focus on statistical comparisons, which are similar to the analyses presented in Sections 4.1 and 4.2 of our work.
> We offer detailed responses to your comments below.
> ### Profiling module:
> The construction of the agents references **CAMEL: Communicative Agents for "Mind" Exploration of Large Language Model Society**. The task design is inspired by **Can Large Language Model Agents Simulate Human Trust Behavior?**.
> ### Participants assignment and CRF:
> Regarding the participant allocation, it was done through random grouping based on psychological experiments, rather than using specific criteria for grouping. After the grouping, we collected and analyzed relevant metrics based on our experiment, as shown in Table 1. In our scenario, Cognitive Reward Functions are different from Cumulative Reward Functions. cumulative reward function(CRF) $R_{i,j}(i, y)$.
> ### Sections 3.2:
> - Regarding payoff and \(E\), we have provided additional details in Appendix \subsection{BREM}.
> - Linear models are easy to construct and interpret, as the weights (\(\beta_1\) and \(\beta_2\)) can directly quantify the impact of each variable on belief updating. In many psychological and behavioral science theories (e.g., the Motive Cocktail framework), individual behavior adjustments (such as belief updating or decision-making) are often considered as linear combinations of prior beliefs, rewards, and environmental factors. Linear models effectively simulate this "cumulative effect" pattern, where specific variables (e.g., beliefs, rewards) exert significant additive or multiplicative impacts on the target variable. Linear models align well with theoretical assumptions in psychology and behavioral science while also offering operational simplicity and interpretability in data analysis, making them a suitable choice for describing belief updating mechanisms.
> - As for emotions, this is one of the key points we are exploring. It is also a common area of investigation in psychology. Here, we aim to investigate whether emotions can be represented by \(T\) in the BT model.
> - For BDF backfires on beliefs, please refer to Equation 8.
> ### Table 2
> Condition1 represents "Selfish" and condition2 represents "Extreme Selfish." The behavior in these conditions is determined by the fairness level of the current environment, cumulative rewards, and current beliefs. Here, the defined "belief" can also be understood as a value system regarding fairness judgments.  When beliefs are stronger, individuals may choose to reject, leading to no rewards. However, in real-world environments, when humans cannot observe varying degrees of unfairness, they tend to compromise and choose to accept (seeking benefits). In contrast, GPT-4o is more rational, and thus, its final score is lower than that of humans.
> - Reference can be made to Figure 4(c)
> ### Section 4.3
> A clear causal relationship can be established between emotion and third-party punishment behavior, as the experimental design manipulates the level of unfairness, with emotion being triggered by the degree of unfairness. The resulting beliefs are thus influenced by the unfairness factors, allowing us to indirectly hypothesize that emotion might also affect beliefs. In our study, we model emotion as the temperature parameter *T* in the BT model to explore this connection. Our findings reveal a stronger correlation between beliefs and actions (as shown in Figure 7), leading us to further infer that emotion may influence beliefs.
>
> We appreciate the reviewer's thorough examination of our work and the identification of typographical errors, which we have corrected. Additionally, we have highlighted these corrections using different colors for clear distinction.

---

> > ### Comment · Reviewer_wqtJ · 2024-11-25
> >
> > I thank the authors for their response. However, I did not find it convincing enough to increase my score.

---

### Official Review · Reviewer_LiRR · 2024-11-04

**Soundness:** 3
**Presentation:** 2
**Contribution:** 3
**Rating:** 5
**Confidence:** 4

**Summary:**

This paper introduces FairMindSim, a simulation framework designed to explore AI alignment by examining the consistency of behavior, emotion, and beliefs in humans and LLMs during ethical dilemmas. FairMindSim employs a modified multi-round classic economic game in which participants, including both LLM agents and humans, encounter unfair reward distributions that simulate moral decision-making scenarios. At the core of this approach is the Belief-Reward Alignment Behavior Evolution Model (BREM), which applies a recursive reward framework to quantify belief evolution in dynamic environments. BREM incorporates decision-making influenced by emotions and beliefs to compare fairness-oriented behaviors between LLM agents (GPT-4o, GPT-3.5, GPT-4 Turbo) and human participants. Findings suggest that GPT-4o demonstrates a stronger response to fairness than humans, whose decisions are more affected by emotional states.

**Strengths:**

1. The experiment was implemented comprehensively and the experimental results were fully analyzed and discussed, providing valuable conclusions for subsequent AI value alignment research.
2. By incorporating social theories into the BREM, the paper offers a novel approach for simulating moral dilemmas and exploring AI value alignment.
3. The study provides a thorough comparison between humans and LLMs in response to unfair scenarios, focusing on emotional responses, beliefs, and social fairness.

**Weaknesses:**

1. It would be better if the authors had a clearer statement of some concepts or experimental settings in their writing, such as condition 1 and condition 2 in the experiment. In addition, the theoretical analysis of the optimal solution in section 3.2 could be better arranged.
2. A large number of typos in the article need to be modified, such as "thedifferences" on line 86, 'EREM' on line 88, "payof f" on line 309, "Let y" on line 317, the second line of formula (7), etc.
3. The analysis of gender and rejection rate of the human group in Section 4.1 is inconsistent with the results in Fig. 4(c).

**Questions:**

1. Could the authors provide a detailed explanation of the level of unfairness $E_j$ in formula (2) in the performed experiments?
2. Although the GPT family is currently the most effective, would adding other large language models make for a more convincing analysis?
3. Would ablation studies (e.g., prompting LLMs to focus on their own utility) strengthen the evidence that LLMs demonstrate altruistic behavior in unfair situations?

---

> ### Author Response · Authors · 2024-11-23
>
> Thank you very much for your careful reading and very insightful comments. We offer detailed responses to your comments below.
> ###  The analysis of gender and rejection rate of the human group in Section 4.1:
> The discrepancy between the results in Table 2 and Figure 3(c) can be attributed to differences in the gender ratios across the experimental groups. Figure 3(c) considers the rejection rate at the individual level, focusing on gender-specific behaviors, whereas Table 2 evaluates the scores at the group level, taking into account overall group performance.
>
> ### Q1:
> Here, $E_j$ represents the degree of unfairness in the *j*-th round of the game. Specifically, it is determined based on the allocation scheme as shown in Figure 2, which describes the allocation between Player 1 and Player 2.
> ### Q2:
> This study focuses on comparing the results of humans and the GPT series models to explore the differences in their performance within the field of psychology. The current experimental design and methodology are based on certain assumptions that require validation through previous related research to ensure scientific rigor and effectiveness. Additionally, the widespread application of GPT series models in the social sciences makes their inclusion in our study a logical choice. Ultimately, we aim for our research to provide interesting experimental results and analyses that contribute to both the AI and psychology fields, fostering the development of interdisciplinary research.
> ### Q3:
> Here, as a supplement to the evidence of altruism in unfair scenarios, we specifically explore a comparison between two unfair schemes. Additionally, in each unfair scheme, every round of the game consists of 20 unfair games.
> We appreciate the reviewer's thorough examination of our work and the identification of typographical errors, which we have corrected. Additionally, we have highlighted these corrections using different colors for clear distinction.

---

> > ### Comment · Reviewer_LiRR · 2024-11-25
> >
> > I thank the authors for the detailed responses. I will maintain the current score.

---

### Official Review · Reviewer_x9UR · 2024-11-05

**Soundness:** 2
**Presentation:** 1
**Contribution:** 2
**Rating:** 3
**Confidence:** 2

**Summary:**

This paper simulates an ecosystem (FairMindSim) for exploring value alignment by using a series of iterated economic games that humans or LLMs participate in.  The chosen economic game is an ultimatum game played between (simulated) Players 1 and 2, and the opportunity for third-party punishment by Player 3 (the participant).  They propose the  Belief-Reward Alignment Behavior Evolution Model (BREM) to explain their findings.


My background does not allow me to comment on the technical aspects of this paper, so I will mostly comment on the high-level potential contributions.

**Strengths:**

Economic games are one of the central tools that behavioral scientists have to understand human judgments and decision-making in highly controllable environments that are amenable to modeling and quantifiable predictions and analysis.  So it is an interesting idea to use them as a tool for measuring value alignment.

**Weaknesses:**

This paper asks a series of ambitious questions, though my worry is that prior work has not yet laid the groundwork for these questions to be asked with rigor and clarity and so the result is that a lot of questions are touched on in a cursory and haphazard way, rather than diving into one question and exploring it thoroughly.

For instance, FairMindSim incorporates information about a user's profile (age, gender, autism spectrum quotient scores, and anxiety scores, to portrait personality and behavior), players' previous decisions and mood states, emotional reactions to observations of ultimatum games, third party punishment, and re-assessment of emotional states following punishment.  Studying the effects of any one of these things could be interesting, but I'm a bit confused about what the use of studying them all simultaneously is, given that we know so little about how each of them operate in this domain.  The authors also don't seem to analyze the effects of many of the elements of FairMindSim, leaving me wondering what elements were important and which weren't.

**Questions:**

I am fundamentally confused about why emotion is such a central topic of focus in this paper.  Should LLMs be aligned with human emotional reactions?  Is there precedent in the behavioral economics literature to focus on emotion and its relationship to third party punishment?

The authors introduce the idea of a "belief" about the environment which can be updated as they gain more information, but I'm having trouble figuring out what a "belief" amounts to in the authors' economic game paradigm.

Small points:
For Figure 3: It would be helpful to explain all abbreviations in the caption.  Figure 3 seems central to understanding the model architecture, but it is very difficult to parse.  Can the authors expand on the caption?  Giving an example of how information flows through the system that is grounded in their experimental paradigm could also be very helpful.

Why did the authors choose to include only unfair offers in FairMindSim?  Do they think this had an impact on the results in some way?  Would things look different if a larger range of offers were used?

---

> ### Author Response · Authors · 2024-11-23
>
> Thank you very much for your careful reading and very insightful comments.
>
> Data on age, gender, Autism Spectrum Quotient (AQ) scores, anxiety scores, and portrait personality traits are collected from human participants to define the agent, enabling the alignment of the LLM's characteristics with those of humans. This approach ensures that the LLM can better mimic human behavior and emotional responses. Ultimately, these human-derived parameters provide a foundation for analyzing the behavior and emotional patterns of the agent, facilitating a deeper understanding of how the LLM interacts in a human-like manner.
>
> We offer detailed responses to your comments below.
>
> #### Q1:
> - In this work, emotions are not considered the core focus. Instead, emotions are hypothesized and simulated using the temperature \(T\) in the BT model. We do not assume that the emotional responses of LLMs are similar to those of humans, which is also a point I hope to further explore.
> - There is relevant literature exploring the relationship between emotions and third-party punishment. Xiao and Houser (2005) investigated how the expression of emotions influences human punishment behavior, highlighting the role of emotions in driving such decisions ([Xiao & Houser, 2005](https://doi.org/10.1073/pnas.0502399102)). Similarly, Raihani and McAuliffe (2012) examined the motivations behind human punishment and argued that it is primarily driven by inequity aversion rather than a desire for reciprocity, further emphasizing the emotional foundations of punitive behavior ([Raihani & McAuliffe, 2012](https://doi.org/10.1098/rsbl.2012.0470)).
>
> #### Q2:
> Regarding the explanation of beliefs. To explore the various socioeconomic motivations, which we refer to as beliefs, that interact to influence individual altruistic behavior.
>
> #### Q3:
> Regarding Figure 3, we have provided additional supplements and updated it accordingly.
>
> #### Q4:
> Here, our FairMindSim focuses on the issue of moral dilemmas, specifically decision-making dilemmas, where unfairness is one of the environmental factors. Of course, other additional factors can also be incorporated by substituting them into the \(E\) variable in our BREM model.
>
> We appreciate the reviewer's thorough examination of our work and the identification of typographical errors, which we have corrected. Additionally, we have highlighted these corrections using different colors for clear distinction.

---

### Meta-Review · Area_Chair_yc8S · 2024-12-17

**Metareview:**

Reviewers liked the framework proposed by the authors and found the observations interesting. However, reviewers are generally unsure to what extent the proposed framework is useful and to what extent the conclusions drawn are trustworthy. These could be due to confusions about the design choices or the quality of presentation. The rebuttal was helpful but did not change the overall sentiment.

**Additional Comments On Reviewer Discussion:**

See above.

---

### Decision · Program_Chairs · 2025-01-22

Reject